

# IoMT based smart healthcare system to control outbreaks of the COVID-19 pandemic

Nouf Abdullah Almujally[1], Turki Aljrees[2], Muhammad Umer[3], Oumaima Saidani[1], Danial Hanif[3], Nihal Abuzinadah[4], Khaled Alnowaiser[5] and Imran Ashraf[6]

[1] Department of Information Systems, College of Computer and Information Sciences, Princess Nourah bint Abdulrahman University, Riyadh, Saudi Arabia

[2] Department College of Computer Science and Engineering, University of Hafr Al-Batin, Hafar Al-Batin, Saudi Arabia

[3] Department of Computer Science & Information Technology, The Islamia University of Bahawalpur, Bahawalpur, Pakistan

[4] Faculty of Computer Science and Information Technology, King Abdulaziz University, Jeddah, Sauid Arabia

[5] Department of Computer Engineering, College of Computer Engineering and Sciences, Prince Sattam Bin Abdulaziz University, Al-Kharj, Saudi Arabia

[6] Information and Communication Engineering, Yeungnam University, Gyeongsan si, Daegu, South Korea

Corresponding authors
Muhammad Umer,
umer.sabir@iub.edu.pk,
umersabir1996@gmail.com
Imran Ashraf, imranashraf@ynu.ac.kr

## ABSTRACT

The COVID-19 pandemic caused millions of infections and deaths globally requiring effective solutions to fight the pandemic. The Internet of Things (IoT) provides data transmission without human intervention and thus mitigates infection chances. A road map is discussed in this study regarding the role of IoT applications to combat COVID-19. In addition, a real-time solution is provided to identify and monitor COVID-19 patients. The proposed framework comprises data collection using IoT-based devices, a health or quarantine center, a data warehouse for artificial intelligence (AI)-based analysis, and healthcare professionals to provide treatment. The efficacy of several machine learning models is also analyzed for the prediction of the severity level of COVID-19 patients using real-time IoT data and a dataset named 'COVID Symptoms Checker'. The proposed ensemble model combines random forest and extra tree classifiers using a soft voting criterion and achieves superior results with a 0.922 accuracy score. The use of IoT applications is found to support medical professionals in investigating the features of the contagious disease and support managing the COVID pandemic more efficiently.

## INTRODUCTION

In March 2020, the World Health Organization (WHO) declared COVID-19 a pandemic. COVID-19 is a ribonucleic acid virus also known as the 'severe acute respiratory syndrome' and the declaration was based on examination and consideration of its severity and spread. As it is a contagious and infectious disease, it can spread between humans and animals.

Major symptoms are dry cough, fever, sore throat, and diarrhea, among others (*Adhikari et al., 2020*). A cost-effective solution is needed to overcome the fast-spreading COVID-19 and reduce the burden on the health system globally. Internet of Things (IoT)-based systems are capable of transmitting data *via* the network and can be accessed from anywhere. IoT can connect things across the globe for information exchange *via* sensors following communication protocols (*Perwej et al., 2019b*).

According to WHO, the most important preventative method is social distancing, in which individuals stay a specified distance from one another, therefore minimizing physical contact with virus carriers. The use of technical instruments to enforce social separation is of primary importance. Artificial intelligence (AI) technologies and approaches may play an important part in tackling the difficulty of applying social distancing. The use of AI technologies can aid in the detection and diagnosis of infections using AI-enabled tools, medical imaging, and IoT techniques. A variety of sophisticated neural network-based solutions have been designed and provided to forecast and monitor the progress of this disease automatically. Furthermore, AI approaches can be used for contact tracking by finding hotspots and clusters. AI has proven itself to play a significant role in providing better preventative and predictive healthcare solutions (*Yadav et al., 2022*).

IoT-connected technology-enabled remote monitoring in healthcare showed the potential for patients to be securely and healthily maintained and empowered clinicians to give superior care. It also increased patient engagement and satisfaction levels by facilitating and expediting communication with providers. Furthermore, remote monitoring minimizes hospital stay time and re-admissions. IoT also has a huge impact on cost reduction and medicinal efficacy as well (*Siripongdee, Pimdee & Tuntiwongwanich, 2020*). IoT has revolutionized people's lives, particularly those of the elderly, by providing continuous health monitoring *via* Wearables like fitness belts and other wirelessly linked gadgets (*Basatneh, Najafi & Armstrong, 2018*). IoT devices also aid in asset management by controlling pharmaceutical stock stocks and monitoring environmental parameters such as humidity and temperature control (*Ma et al., 2020*).

IoT-based system can link patients to health service providers in emergency cases. With the use of communication channels, IoT provides advanced services and applications (*Akhtar, Parwej & Perwej, 2017*). From its surrounding environment, IoT devices capture information and send it to the internet or other devices (*Perwej et al., 2019a*). These devices are suitable when the same network is considered; also, they acquire the characteristics for helping with the fight against the COVID pandemic. IoT applications, with a broad range, can help assure that the COVID-related guidelines provided by health officials are being followed (*Pham et al., 2020*). During the COVID breakout, IoT helped by providing vital information with the usage of sensors (*Allam & Jones, 2020*). A robust and efficient IoT-based system is required to cope with the quick spreading of the COVID outbreak.

In this article, a monitoring system for COVID detection is proposed which starts by using its IoT-based sensors for collecting real-time data on an infected person. As systems have IoT-based infrastructure, so early treatment with extra care for severe cases is possible. Cases of recovery can be followed up on. Relevant data collecting on a cloud platform can

also help to better understand the nature of the condition. The proposed system comprises four major components including collecting data for disease, quarantine and health centers, data warehouse, and health officials. The primary goals for the proposed framework are

- To categorize the patients with the severity of the disease and deliver a quick response to lower the death rate.
- Improving the prediction of medical IoT (MIoT) using an ensemble learning model that combines extra tree classifier (ETC) and random forest (RF) for COVID-19 severity prediction.
- Using real-time symptoms-based data, to accurately identify the patients suffering from COVID disease.
- Helping medical professionals for early diagnosis and treatment of infectious diseases.

This article is categorized as follows. Background information is provided in 'Literature Review' while the applications of IoT at various stages of COVID are reviewed in 'COVID Phases and Usage of IoT Devices'. IoT-related challenges are discussed in 'Challenges in IoT'. 'Proposed Framework' provides details about the proposed framework. 'Materials & Methods used for Estimation of Severity of COVID' focuses on the methodology and materials used for experimentation with discussed in 'Results & Discussion'. Lastly, 'Conclusion and future work' provides the conclusion.

## LITERATURE REVIEW

In 2003, the word 'IoT' was devised by Kevin Ashton and got publicized later. After investigating the domain and its different aspects regarding future roles and progress, various companies invested at regular intervals (*Atzori, Iera & Morabito, 2010*). Since IoT is an open network, considering the situation and environment it can connect, organize, share, and react (*Kamdar, Sharma & Nayak, 2016*). Extensive research works have been performed for the exploration of IoT and its application related to business, health, management, and industry. To enable things to have a sense of smell, hearing, and seeing and information sharing is a major area of concern in research. This allows to use of the IoT-based system for monitoring and using them for decision making (*Korade, Kotak & Durafe, 2019*). *Wang et al. (2020)* claim that in the COVID outbreak, after the development of a vaccine, the reachability of patients is another important issue. According to *Toma (2021)*, the most powerful tool for cloud computing, AI and data analytics is IoT. These devices can assist in monitoring, controlling, and disease diagnosis by using their characteristics of recording and data transmission *via* the cloud. Similarly, *Singh et al. (2020)* states that given the current COVID outbreak, facilities based on IoT should be provided to reduce the impact. Due to the increasing number of patients, they can be reached easily by using IoT-based tools to facilitate them with extra care. The IoT-based tools which use sensory products are suitable in such scenarios.

The authors (*Rahman et al., 2020*) presented a health examining system using IoT combined with IoT sensor devices, and health testing using cloud and AI-based tools. Both supervised and unsupervised machine learning models were used on social media and data related to health. With the use of modern AI, cloud computing, speech recognition,

blockchain, and automation, it was possible to enable the monitoring system between patients and doctors. IoT provides secure chat and telehealth features to the monitoring system of health. Such features are offered in smartphone applications with application programming interface (API) and edge computing along with user-friendly interfaces.

*Maghded et al. (2020)* proposed a framework for the detection of COVID that consists of smartphone built-in sensors. The framework is a low-cost solution as smartphones are widely used by the masses for various daily life routine chores. Considering the large computing capacity of smartphones with various sensors like thermo-sensors, hygrometers, color sensors, proximity, wireless sensors, gyroscope, and accelerometers, their proposed AI-based framework reads data from sensors of smartphones and makes predictions about the disease.

The study by *Sujath, Chatterjee & Hassanien (2020)* created a model that may be beneficial in forecasting the outbreak of COVID-19. The authors used various regression methods on the COVID-19 dataset to forecast the disease and the rate of COVID-2019 instances in India. *Yadav et al. (2022)* proposed a real-time crowd-monitoring system to control COVID-19. An object detection technique is applied to detect social distance in public places. Considering the limitations of other methodologies, a unique real-time solution is offered to detect and monitor COVID-19 patients, and the importance of IoT applications in combating COVID-19 is also highlighted.

## COVID PHASES AND USAGE OF IOT DEVICES

After the declaration of COVID as a pandemic by WHO, various attempts and efforts to control its spread and find its cure have been carried out. As asymptomatic and symptomatic types of COVID infections are being seen and dealt with by patients, a global monitoring system is the need of time for the observation of conditions and patients.

To deal and cope with the COVID outbreak at all stages, and for obtaining real-time data about affected persons, IoT is an innovative and scientific platform (*Qi et al., 2017*). It can provide a secure and helpful platform to transmit data by sensors, particularly in large numbers, which are interlinked *via* a dedicated network. Various stages of the COVID pandemic are shown in Fig. 1. By providing tracking and alerting COVID patients, it improves their safety. IoT applications during various phases of the COVID pandemic are discussed below.

### Phase I: infection and disease detection

In the first step, health-related data of COVID patients from various locations is captured and managed using IoT. Early diagnosis of infection is highly needed for the prevention of disease spread, specifically when asymptomatic patients are concerned. This would help in controlling COVID spread as after detection early, patients can be isolated and treated accordingly (*Sabeti, 2020*). IoT devices can play a significant role in capturing the temperatures as samples for laboratory tests, which makes early diagnosis possible thereby allowing health professionals to devise plans for treatments and save lives. It involves understanding the symptoms like body aches, taste and smell loss, sore throat, cough, vomiting, fever, and diarrhea (*Khajenoori et al., 2020*). During the infection of COVID,

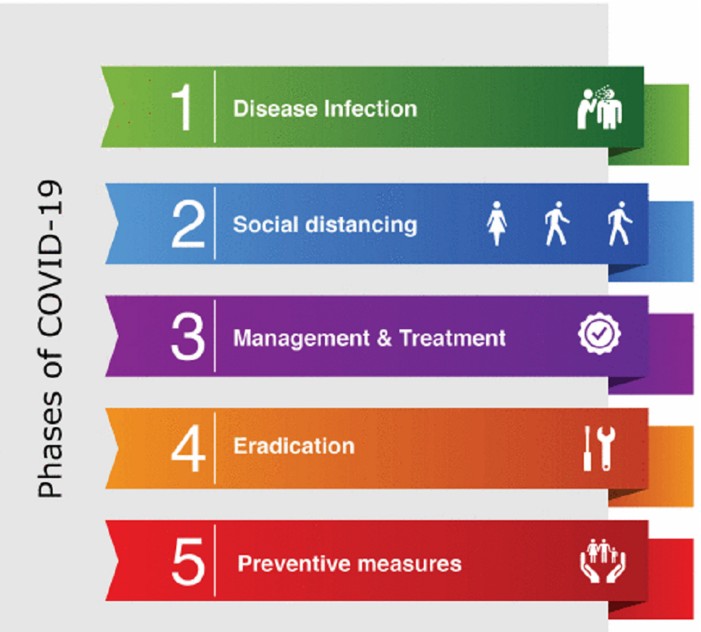

**Figure 1   Phases of COVID.**

fever is the most common symptom, as it may exceed 100 Fahrenheit (*Rahman et al., 2020*). Thus, during this phase, IoT can be used for obtaining information.

### Smart thermometers

Several IoT-based thermometers are available nowadays for recording body temperature and are available in various forms like patch, touch, and radiometric and are extensively used for the immediate detection of diseases (*Liu et al., 2020*). Unlike classical thermometers, IoT-based thermometers can be paired with mobile applications allowing them to track and show records. It enables a more effective reading in the pandemic situations. For example, the Kinsa thermometer delivers temperature with an app connectivity for tracking location and medication logs and is being used in the US for tracing the COVID suspected area (*Chamberlain et al., 2020*). VICOODA, iHealth Forehead, i-fever, and i-sense are smart thermometers and are wearable as they can stick to the body, and their use can speed up the diagnosing process.

### Smart glasses

Smart glasses based on IoT are considered the best, given the current scenario of the COVID pandemic to reduce human interaction as they use optical and thermal cameras for monitoring the crowds and are later sent to health officers on smartphones (*Abdulrazaq et al., 2020*). Smart glasses based on infrared sensors with a capacity of monitoring up to 200 people at a time are being produced in China (*Bright & Liao, 2020*). Vuzix is another example of smart glasses with a thermal camera for temperature recording and updating health officials in real time (*Wilkins, 2020*).

### Smart drones

For monitoring the COVID-affected areas and patients, IoT-based drones are also helpful as they can be used for monitoring the areas effectively. Such deployments can lessen human interaction and can go to difficult-to-reach areas and places taking less time (*Chamola et al., 2020*). Thermal drones can be used for monitoring crowds and can allow identifying people with a high fever. Similarly, if thermal guns are used, recording live streaming and temperature is possible quickly. With the usage of sensors, the Canadian company has developed a drone to monitor the temperature in remote locations by monitoring sneezing, coughing, and heart rate for the detection of affected people (*Kumar et al., 2020*).

### Autonomous swab testing robots

Health officials are using robots based on IoT for the detection and diagnosis of COVID-19. It reduces chances of infection and mental stress (*Yang et al., 2020*). Autonomous robots can be used in the first phase for swab samples from patients for infection detection, thus providing lab experts less interaction with patients. There is another intelligent robot that allows scanning patients for temperatures and other symptoms from a 1-meter distance within 10 s (*Navi, 2020*).

## Phase II quarantine and social distancing

Social distancing and quarantine are needed after the identification of contagious disease to prevent it from being spread with patients still required to be monitored at home or hospitals. Quarantine is recommended for both confirmed and suspected cases. Lockdown which is the sealing of areas of a city or whole country for a certain period for the sake of disease control can be used (*Kuhbandner et al., 2020*). The main goal is to prevent the disease from being transmitted to patients. In this phase, IoT devices can be used for monitoring patients (*Wilder-Smith & Freedman, 2020*).

### IoT Q-bands

For tracking COVID cases, IoT-based Q-bands are another suitable option. It provides cost-effectiveness and allows pairing them with smartphones using Bluetooth. Authorities can monitor the quarantined patients wearing bands with being interacting with patients. The band also comes with the feature of location sharing every two minutes so it alerts the authorities if someone tries to remove the band or tries to leave the location. E-bracelets were used on ankles in the US during the duration of the quarantine of patients (*Nasajpour et al., 2020*). In Hong Kong, e-wristband paired with a smartphone were used for patient monitoring at airports (*Hui, 2020*).

### IoT buttons

A big problem during the COVID pandemic was controlling the spread of the virus, so in Canada, IoT-based buttons were used and placed for alerting the authorities regarding patients' families and suspicious areas for implementation of emergencies (*Udgata & Suryadevara, 2020*). These are battery-operated and can be placed anywhere according to the requirements (*Borkar, 2020*).

### Easy bands

This device works by sensing and collecting data and works within a certain range by showing alerts by using LED lights with its working varying with persons. Other bands start showing alerts, if someone with a red LED comes in the range, thus, allowing to trace suspected cases with maintaining social distancing. It is considered a low-cost solution as compared to smartphones (*Hussein et al., 2020*).

### Smart helmet

For maintaining social distance and reducing human contact, wearable helmets based on IoT are another solution (*Mohammed et al., 2020*). Upon detection of high temperature by helmet or thermal camera, a picture of a suspected person is sent to the attached device along with a warning. To monitor the situation during the quarantine (*Ghosh, 2020*), these devices were implemented in Italy, UAE, and China. KCN901 is an example of such a device that can provide a 96% accuracy (*Ghosh, 2020*).

### Proximity trace

These devices are used in industrial areas for maintaining social distancing between workers (*Reelfs, Hohlfeld & Poese, 2020*). As it is wearable and can be placed on the hat or the body of the worker, it produces sound upon closing a worker with another worker and allows workers in putting more focus on work rather than being worried about another colleague.

### Broadcasting drones

These drones were used in many countries during quarantine and lockdown to ensure that people stay inside. In addition, they are reported to be used to monitor social distancing, and to approach the area with limited or no internet (*Couch, Robinson & Komesaroff, 2020*). An example is the "Go home" broadcast message used in Spain (*Singla, 2020*).

### Social robots

These robots are designed to keep the patients accompanied by communicating with them for reducing fatigue, boredom, and stress, as mental health problems were raised during quarantine sessions. An example is Paro which is developed for reducing and relieving the stress of patients in quarantine (*Odekerken-Schröder et al., 2020*).

## Phase III management & treatment
### Telemedicine

In remote areas, where physicians are not available considering the various factors, this service is getting popular as it allows the monitoring of diabetes, fever, heart rate, and other indications remotely without the need for physical contact. Data is captured using sensors and later uploaded to the cloud where doctors can monitor the details of patients with laptops or smartphones. During the COVID pandemic, it played a crucial role. Healthcare professionals are monitoring the patients, which are using IoT-based devices for healthcare. An example of such services is Sehatyab (*Nazir, 2020*), which is a leading service to provide continuous care (*Care, 2020*).

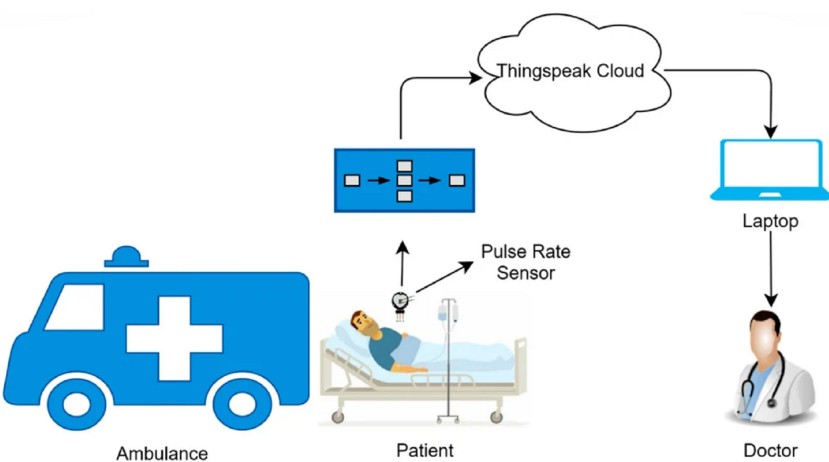

**Figure 2** **IoT-based ambulance** (*Bajaj, Kumar & Kaushal, 2022*).

### IoT-based ambulance

During the outbreak of COVID, respiratory tract blockage was the major cause of the increased number of deaths. With such patients requiring intensive care, the medical staff involved with such infected patients remained under high stress. Ambulances based on IoT assisted medical professionals with patients in critical situations (*Vistro et al., 2020*). For providing fast interaction among actors, a high-speed vehicular network is a must as it allows for attaining the full advantage of these ambulances. Medical professionals access the information of patients remotely by using IoT. Figure 2 shows an image of an IoT-based ambulance.

### Social media

Because social media is being used by millions of people, it makes it suitable for connecting people regarding their health. To deal with the COVID pandemic, patients can communicate with physicians and can also attend their live sessions. Resultantly, the number of the hospital for treatment is reduced as there is no need for physical contact for health professionals for giving medical prescriptions to patients (*Machado et al., 2020*).

### Phase IV eradication

Implementations of lockdown whether it is citywide or countrywide has caused economic problems and in the current phase of recovery, crucial actions are being taken by governments for its eradication to the full extent. As institutions and businesses are re-opening in phases, a significant role is also played by IoT for the complete removal of viruses.

#### *Disinfectant spraying*

To get rid of the virus, it is very important to keep the area disinfected and sanitized. Public areas can be sprayed before opening with IoT-based smart devices. Drones are used

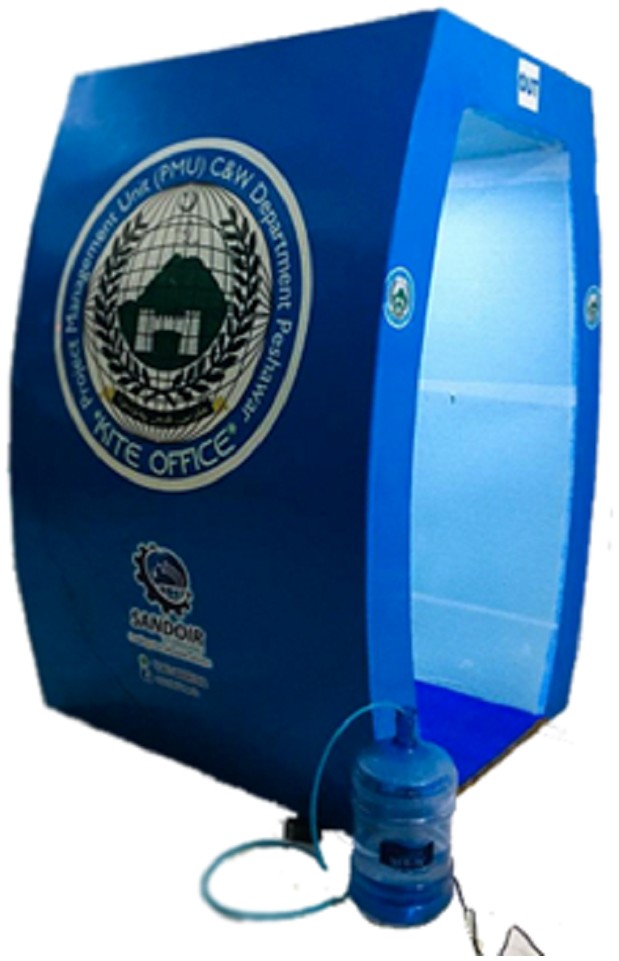

**Figure 3** Automatic sanitizing gate by Sandoir Technologies (*Technologies, 2020*).

for spraying and sanitizing a hundred-meter area in an hour. It is also used by Spain for sanitization (*Kumar et al., 2020*).

### Automatic sanitizing spray

To protect people against COVID infection, an epidemic smart tunnel that made use of an automatic sanitizing spray system was proposed (*Pandya, Sur & Kotecha, 2020*). The proposed system works by detecting humans and their motion thereby disinfectant spraying whenever someone passes through it. Figure 3 shows a sanitizing door designed by a Pakistani electronic corporation Sandoir Technologies.

## Phase V preventive measures

Government-provided preventive measures for the people are listed as follows. These measures vary from country to country and city to city within a country concerning the virus's spread.

1. Use of sanitizers and washing hands with soaps in case of contact with people,

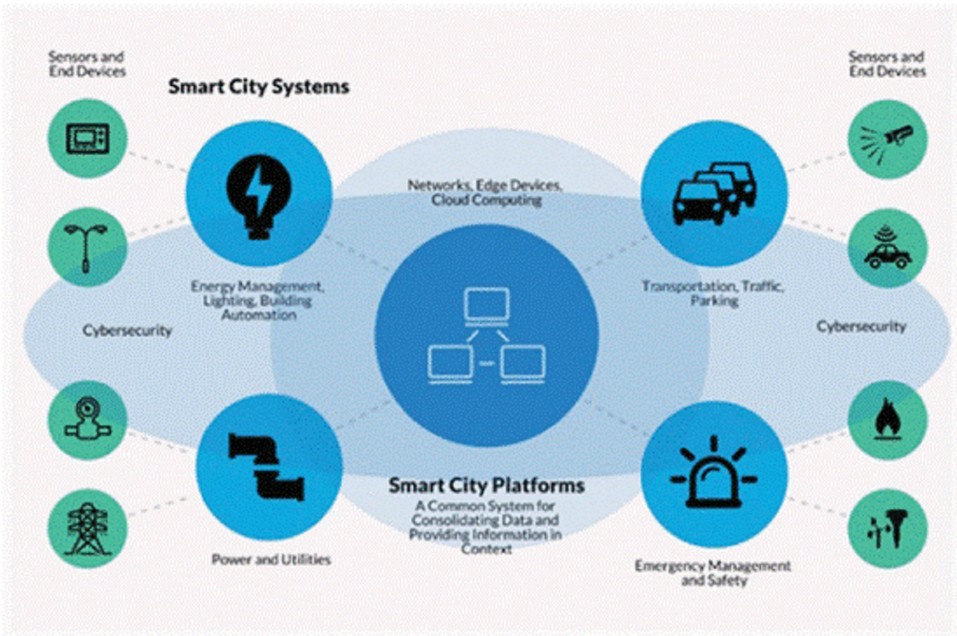

**Figure 4** Smart city concept (*O'Brien, 2020*).

2. Maintenance of social distancing,
3. Pregnant women and old people must stay at home,
4. Touching eyes, ears, mouth, and nose unnecessarily, should be avoided,
5. Floors and handles of doors must be sterilized or sanitized regularly.

### Smart city

Due to the COVID outbreak, IoT devices for monitoring the important places of the city like hospitals, airports, bus terminals, markets, and subways are installed. In cities where data related to arrival and departure time is being monitored and analyzed, this concept of a smart city is used. Li (*Li, Batty & Goodchild, 2020*) studied various such sensors which are installed for providing real-time information at different places. For activity monitoring, authorities are using wearable IoT devices. Figure 4 shows an example smart city.

## CHALLENGES IN IoT

### Scalability

As IoT and devices based on IoT began to grow considerably along with the advancement of technology, scalability is a major challenge. Alongside their use in houses, these devices are used for preventing COVID spread. For tracing the patients of COVID and assessment of social distancing implementation, BLE wearables are used, with their data management, and stored on the cloud and so there is a need for powerful computational systems, and large storage is required. Also with scalability, energy needs are heightened.

### Reduced bandwidth and erroneous data

With the outbreak of COVID, the use of IoT-based devices is growing. There are many sensors in IoT-based wearables for controlling the spread of COVID with each sensor sending information by using APIs. Cellular networks are providing bandwidths for the transfer of data between sensors of IoT and clouds but after increased usage of wearable IoT devices, the offered bandwidth is inadequate for transferring real-time data transfer. It sometimes results in the transfer of erroneous data to the cloud, which disturbs the working and efficiency of wearable IoT.

### Miscommunication & privacy issues of data

Considering the previously reviewed bandwidth and scalability issues for wearable IoT devices, they are causing other issues regarding privacy and data miscommunication. The consequences of these problems are

1. There is the live transmission of data along with personal information from IoT-based devices which are attached to people. As there is a stability problem in IoT, for the secure transmission of data between the device and the cloud, there are no cryptographic algorithms available. So, data interception on communication channels is possible resulting in leakage of personal and other sensitive information (*Kamal & Tariq, 2018*; *Aman, Basheer & Sikdar, 2019*).

2. The access to the personal data of the patients should be prioritized and provided only to the relevant persons.

### Interoperability issues

Interoperability is a big concern for the proper functioning of IoMT, after bandwidth, privacy, and scalability issues. For accurately transmitting real-time data, a dynamic scheme is needed. Health officials should be provided training sessions in this regard. During COVID, additional goals should be set for underdeveloped and remote areas (*Gupta, Christie & Manjula, 2017*).

## PROPOSED FRAMEWORK

In this section, we elaborate on the suggested framework based on IoT which can be applied for real-time monitoring and detection of coronavirus cases. The proposed framework can assist with the understanding of the different phases of COVID and its treatment. Figure 5 shows the architecture of the proposed framework. It comprises four phases that are interlinked by the cloud.

### Collection of data for symptoms of disease

This module uses IoT-based sensors for collecting COVID symptoms including fever, coughing, sore throat, tiredness, and shortness of breath in real-time. Various biosensors are available in the market, for example for the detection of cough, audio sensors with aerodynamics, fever detection temperature sensors, and oxygen-based sensors can be used for shortness of breath. Similarly, images can be used for sore throat detection and chest pain and fatigue can be detected with heart rate sensors.

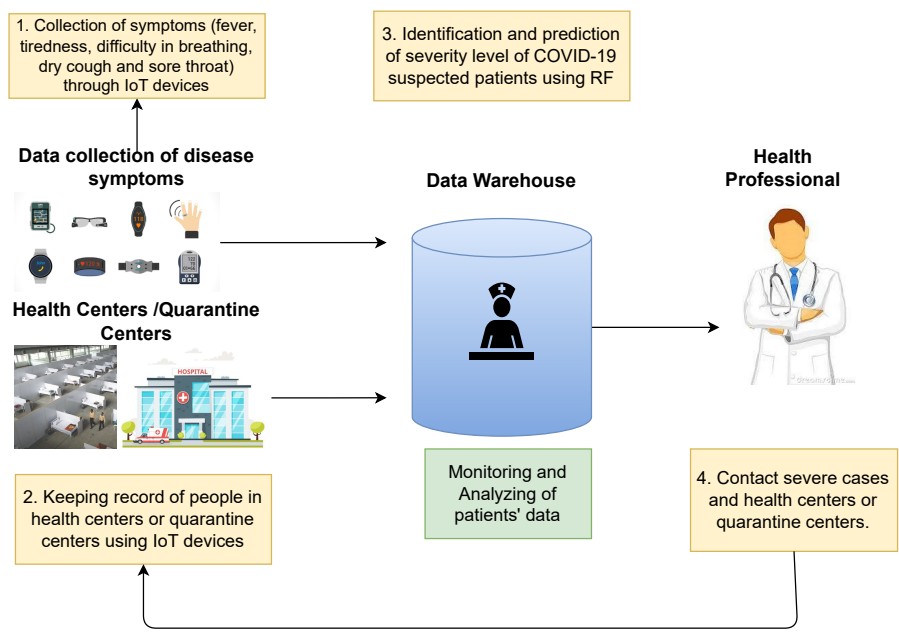

**Figure 5  Flowchart of the proposed framework.**

## Quarantine and heath centers

The goal of this component is to keep track of the information on quarantined or isolated patients by recording technical and non-technical data. Non-technical data includes last week's information like travel and illness history, gender, and age while technical data contains symptoms and the patient's response in the course of treatment. During treatment, both records are vital to deal with.

## Data warehouse

Machine learning algorithms are used for analyzing the data which is uploaded by sensors to data centers and deliver helpful information on real-time data processing. Detection and treatment of COVID are achievable with the help of these models. Researchers can understand the nature of the disease by comprehensive analysis of these results.

## Health professionals

Health professionals make use of results generated by machine learning models for dealing with cases of COVID and suspected patients getting quick responses and being screened for confirmation which results in identified patients getting treatment on time. The Internet is used for connecting components of the framework. Sensors upload real-time data of the patients to the servers. After the maintenance of records, results generated by the machine learning models are discussed with physicians and health officials. Additionally, to stop the further spread of the virus, and provide precautions to the public, the proposed framework can also be used.

**Table 1  Descritpion of the dataset attributes.**

| Attributes | Description |
|---|---|
| Country | List of countries person visited. |
| Age | Classification of the age group for each person, based on WHO Age Group Standard |
| Symptoms | According to WHO, 5 are major symptoms of COVID, Fever, Tiredness, Difficulty in breathing, Dry cough, and sore throat. |
| Experience any other symptoms | Pains, Nasal Congestion, Runny Nose, Diarrhea, and Other. |
| Severity | The level of severity, None, Mild, Moderate, Severe |
| Contact | Has the person contacted some other COVID Patient |

# MATERIALS & METHODS USED FOR ESTIMATION OF SEVERITY OF COVID

In this section, machine learning models and datasets used in the data warehouse are discussed. Experiments are carried out for identifying the COVID infection quickly.

## Data set

A dataset named "COVID Symptoms Checker" is available (https://www.kaggle.com/iamhungundji/covid19-symptoms-checker) and contains features for the classification of whether a person is infected or not. These features are based on indications set by WHO. There are twenty-seven attributes like age, country, symptoms, contact, and severity in the dataset further categorized into five groups. Table 1 details of attributes. In this research, 26 attributes are used for the prediction of the severity level of persons infected with COVID Fig. 6. There are 316,800 records in the dataset and there are 79,200 records having severity associated with it.

## Data preprocessing

There are many attributes in the dataset with unique values, like "none", "mild", "moderate", and "severe" which are possible values for severity attributes. There are binary values in "Yes" and "No" format in tiredness, dry cough, fever, sore throat, and difficulty in breathing attributes. Since vector numeric form is required for learning classifiers, so label encoding is applied for each attribute for conversion of text values into respective numeric.

## Predictive models

The primary idea of the machine learning models is to assess the present condition of the patient. For assessment and prediction of the severity of disease level in patients infected with coronavirus, a particular dataset is used. The models used in this study are implemented in Python and *Scikit-learn* (*Pedregosa et al., 2011*) and the performance of these models is compared.

Random forest (RF) makes use of decision trees and generates trees for reducing the variance. It is equally good for classification and regression problems. The bagging

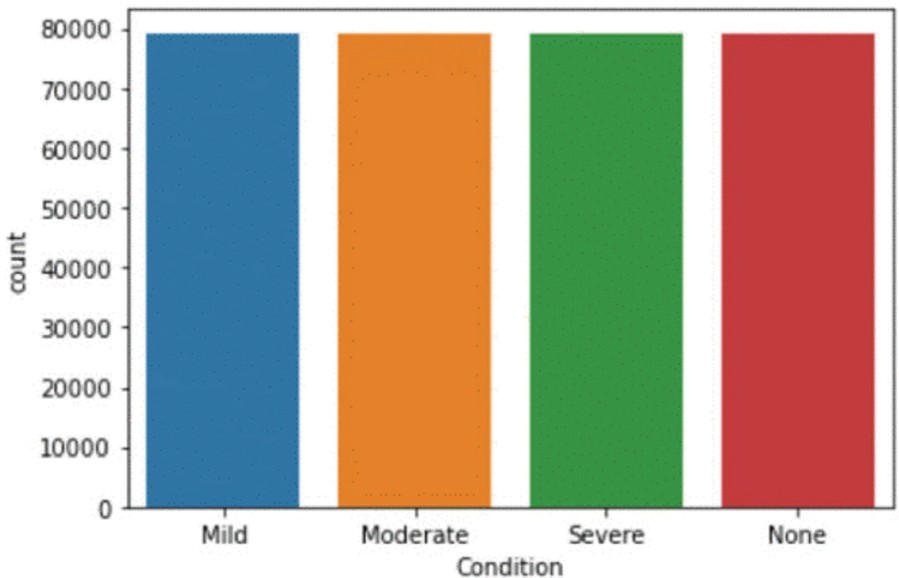

**Figure 6** **Visual representation of the COVID symptoms checker dataset.**

technique is used to make the final prediction using the majority voting from tree results. To escape variance and overfitting, a random subset of features and dataset is used (*Breiman, 2001*).

AdaBoost (AB) is a boosting procedure and provides outstanding performance for classification problems. It learns from misclassified instances appearing in previous iterations and then transforms the weak learners into strong learners based on its results (*Schapire & Singer, 1999*). The main idea here is to get better performance with high accuracy by combining with weak learners. AB makes use of feature scores for finding the important features.

Gradient boosting machine (GBM) is a collaborative technique based on boosting and is used in classification and regression. This technique combines weak learners like decision trees for final prediction (*Natekin & Knoll, 2013*). Boosting model converts weak learners into strong learners with each iteration. For performance improvement, the loss function is used which helps in finding the effectiveness of the coefficients of models over underlying data.

For generating and aggregating results from multiple decision trees, the extra tree (ET) model is used, which makes final predictions. It does so by averaging the decision with various samples of the dataset (*Geurts, Ernst & Wehenkel, 2006*). Rather than picking the best node, an extremely randomized tree utilizes the whole sample from the dataset and picks the random root node. It is also considered an extremely randomized tree classifier for this reason.

The voting classifier (VC) model uses the probability scores from various base learners and predicts the output. In these types of models, the output can be acquired using two ways

1. Soft voting
2. Hard voting

In soft voting, an average of every model is calculated and then the output value with high probability is chosen whereas, in hard voting, the majority of votes from classifiers are used for the final output. In the current study, soft voting is used and SGD and LR are combined for VC. LR is a linear method that utilizes probabilities for making predictions. For modeling dependent variables, it uses a logistic function (*Wright, 1995*). It can also be used for classification and regression. Taking derivatives from the training date is gradient descent, while SGD is used for taking derivatives from every entry of training data and was introduced as a technique for variance reduction (*Johnson & Zhang, 2013*). For classification tasks, SGD and LR are well-known techniques.

## Proposed methodology

In state-of-the-art research works, ensemble learning techniques are considered to get more reliable results than individual models. Ensemble learning benefits from the wisdom of the crowd, leveraging the collective knowledge and predictions of multiple models to improve accuracy, robustness, and generalization. These are the reasons that proposed ensemble learning models predict the severity of COVID patients more accurately.

The workflow for predicting the severity of COVID patients is depicted in Fig. 7. The model is a combination of two machine learning models, RF and an extra tree classifier (ETC). Experiments are conducted using the COVID severity prediction dataset. The proposed model is applied to the dataset. The data is split with a ratio of 0.7 to 0.3, where 70% data is used for training and 30% for testing. The performance of the model is evaluated using precision, recall, and F-score metrics.

In this work, ETC and RF models are combined using the soft voting criteria. The final output is decided by using the probability of each class. Algorithm 1 shows how the proposed ensemble model works.

Mathematically, the soft voting criteria can be represented as

$$\widehat{p} = argmax \sum_{i}^{n} RF_i, \sum_{i}^{n} ETC_i \tag{1}$$

where the probability values against the test sample are denoted by $\sum_{i}^{n} RF_i$ and $\sum_{i}^{n} ETC_i$ and $\widehat{p}$ shows the final prediction. The probability values for each instance using ETC and RF are then passed through based on soft voting.

Each test sample is fed into ETC and RF models for label prediction. These models assign a probability score to each class of the given sample. For example, if the RF model's probability values are 0.5, 0.6, 0.7, and 0.8 for four classes, respectively, and the ETC model's probability values are 0.6, 0.7, 0.8, and 0.9 for four classes, respectively, and $P(x)$ represents the probability value of $x$ ranging from 0 to 1, the final probability is determined as

$$P(1) = (0.5 + 0.6)/2 = 0.55$$
$$P(2) = (0.6 + 0.7)/2 = 0.65$$
$$P(3) = (0.7 + 0.8)/2 = 0.75$$

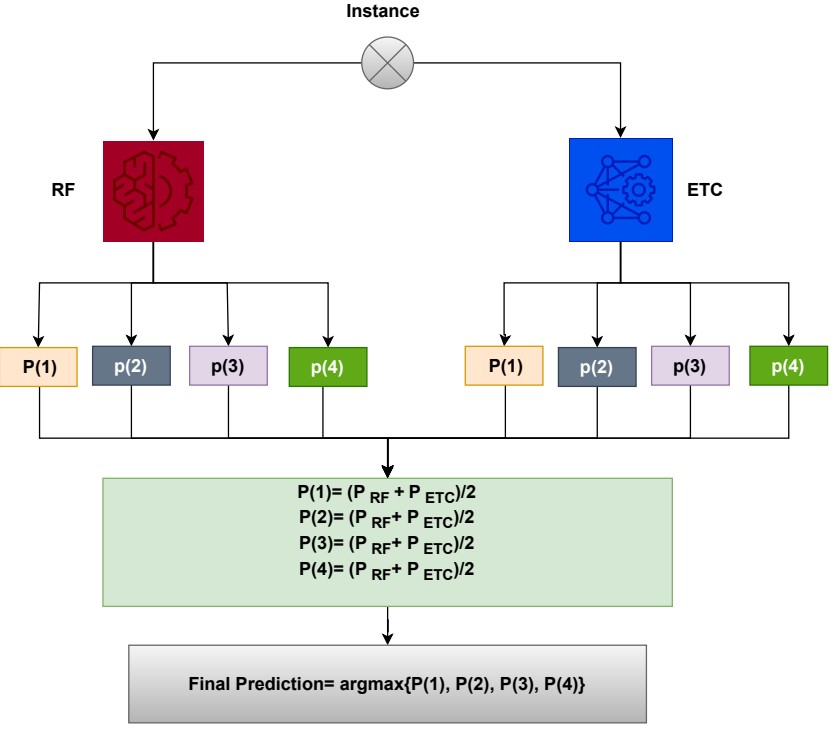

**Figure 7** **Workflow diagram of the proposed voting classifier (RF+ETC) model.**

---

**Algorithm 1** Ensembling ETC and RF models.

---

**Input:** input data $(x,y)_{i=1}^N$

$M_{RF}$ = Trained RF

$M_{ETC}$ = Trained ETC

   **for** $i = 1$ to $M$ **do**

     **if** $M_{RF} \neq 0$ & $M_{ETC} \neq 0$ & $training\_set \neq 0$ **then**

      $P_{RF_1} = M_{RF_1}.probability(class1)$

      $P_{RF_2} = M_{RF_2}.probability(class2)$

      $P_{RF_3} = M_{RF_3}.probability(class3)$

      $P_{RF_4} = M_{RF_4}.probability(class4)$

      $P_{ETC_1} = M_{ETC_1}.probability(class1)$

      $P_{ETC_2} = M_{ETC_2}.probability(class2)$

      $P_{ETC_3} = M_{ETC_3}.probability(class3)$

      $P_{ETC_4} = M_{ETC_4}.probability(class4)$

      Decision function $= max(\frac{1}{n}\sum_{classifier}(Avg(P_{RF_1},P_{ETC_1}),$

      $Avg(P_{RF_2},P_{ETC_2}),Avg(P_{RF_3},P_{ETC_3},Avg(P_{RF_4},P_{ETC_4})$

     **end if**

     return final label $\widehat{p}$

   **end for**

---

**Table 2  Performance of machine learning models.**

| Classifiers | Accuracy | Precision | Recall | F-score |
|---|---|---|---|---|
| AdaBoost classifier | 0.604 | 0.712 | 0.781 | 0.746 |
| Gradient boosting classifier | 0.671 | 0.782 | 0.771 | 0.776 |
| Extra tree classifier | 0.697 | 0.748 | 0.784 | 0.766 |
| Random forest classifier | 0.754 | 0.794 | 0.810 | 0.802 |
| Voting classifier (AB+GBM) | 0.826 | 0.834 | 0.850 | 0.842 |
| Voting classifier (AB+ETC) | 0.844 | 0.852 | 0.869 | 0.851 |
| Voting classifier (AB+RF) | 0.851 | 0.872 | 0.876 | 0.874 |
| Voting classifier (ETC+GBM) | 0.852 | 0.872 | 0.896 | 0.882 |
| Voting classifier (RF+GBM) | 0.892 | 0.902 | 0.909 | 0.905 |
| Voting classifier (RF+ETC) | 0.922 | 0.952 | 0.969 | 0.960 |

$$P(4) = (0.8 + 0.9)/2 = 0.85$$

The final prediction will be 4 because it has the highest probability. VC(RF+ETC) chooses the final class based on a class's maximum average probability and combines the predicted probabilities of both classifiers.

# RESULTS & DISCUSSION

In Table 2, a comparison of various machine learning models is shown with precision, accuracy, recall, and F score of all classifiers. According to the given results, models applied in research are effective with the prediction of severity in COVID patients and ETC and RF show the highest performance with 0.922, 0.952, 0.969, and 0.960 scores for accuracy, precision, recall, and F-score, respectively.

However, it is concluded from the results that for the prediction of the severity of COVID patients, all the models based on trees can be feasible and more effective. Several machine learning models are studied in this study and it is found that for the analysis of data, ETC and RF ensembles are the most suitable. They utilize a small number of parameters for tweaking and handling the multi-dimensional data in the better possible way. In the proposed framework, the said model is integrated into the data warehouse part, where a collection of data is done by sensors thereby uploading it to data centers. It will help in decision-making to prevent the COVID outbreak. As per the results, the proposed system based on IoT can make use of RF for the classification of the severity of COVID patients.

## Comparison with state-of-the-art techniques

The performance of the proposed model is compared to that of classifiers employed in *Hassan, Rashid & Hamarashid (2021)* and *Rochmawati et al. (2020)* applied to the COVID symptoms severity dataset. Table 3 presents the results of all models, which reveals that the Voting classifier has shown superior performance. In *Hassan, Rashid & Hamarashid (2021)* iECA has shown the highest accuracy of 0.909. *Saengamnatdej, Molee & Warnnissorn (2023)* observe the effect of SMOTE data upsampling with extreme gradient boosting (XGB) model to predict COVID-19 with an accuracy of 0.812. The research work (*Giotta et al., 2022*; *Mahesh et al., 2022*) implemented tree base learning models and get an accuracy of

**Table 3  Performance comparison of the proposed model with state-of-the-art approaches.**

| Reference | Model name | Accuracy |
| --- | --- | --- |
| *Rochmawati et al. (2020)* | J48 algorithm | 0.835 |
| *Rochmawati et al. (2020)* | Hoeffding Tree | 0.831 |
| *Hassan, Rashid & Hamarashid (2021)* | KNN | 0.890 |
| *Hassan, Rashid & Hamarashid (2021)* | ANN | 0.870 |
| *Hassan, Rashid & Hamarashid (2021)* | SVM | 0.740 |
| *Hassan, Rashid & Hamarashid (2021)* | LVQ | 0.860 |
| *Hassan, Rashid & Hamarashid (2021)* | DeepKNN | 0.895 |
| *Hassan, Rashid & Hamarashid (2021)* | GENClUST++ | 0.865 |
| *Hassan, Rashid & Hamarashid (2021)* | ECA Star | 0.899 |
| *Hassan, Rashid & Hamarashid (2021)* | iECA Star | 0.909 |
| *Giotta et al. (2022)* | Decision Tree | 0.759 |
| *Mahesh et al. (2022)* | ML models | 0.825 |
| *Saengamnatdej, Molee & Warnnissorn (2023)* | SMOTE+XGBTree | 0.812 |
| Current work | **Voting Classifier (RF+ETC)** | **0.922** |

Notes.
The bold value indicates the highest accuracy.

0.759 and 0.825 respectively. On the other hand, the voting classifier of the current study has shown a 0.922 Accuracy score.

## Importance of using IoT

IoT-based devices are and will be contributing significantly to managing the COVID pandemic. These devices do so by real-time monitoring of patients and their data for the sake of locating and identifying the patients. It can also detect fraudulent information from its sources.

Whenever the spread of a pandemic or epidemic is concerned, early identification and tracing of infected people and their isolation are crucial steps. The use of IoT-based devices as a solution for these challenges is getting wide attention. This research highlights the uses of wearable IoT-based solutions for fighting the COVID outbreak. The main goal is tracing the infected persons, spread elimination, and reducing the large impact of the disease.

Using IoT properly will surely aid in the early identification of infected people and making decisions accordingly. IoT-based tools can be applied for capturing data, predicting situations, and enabling government and health officials for making appropriate policies during the pandemic situation.

## CONCLUSION AND FUTURE WORK

By using IoT, devices in the hospital and other locations can be connected *via* a network for controlling the COVID outbreak. These devices can assist doctors, patients, and the health system in identifying patients early thereby managing positive cases in an efficient and better way, globally. A framework based on IoT is presented in the article, for limiting the epidemic of this infectious disease. In the presented framework, data from health centers and patients is collected and then saved in data warehouses for further analysis. Then

machine learning models are used for the prediction of the severity of the disease. The framework also transfers the generated results to health officials for a quick assessment of treatment or isolation of the patients. Several machine learning-based techniques are used on real-time data of the patients. As per outcomes, ETC and RF achieve high accuracy for the identification of the severity of COVID patients. This study discusses the use of IoT devices for monitoring patients during various phases of COVID. The suggested framework can assist patients, health officials, and administration in investigating infectious diseases. The framework can be deployed for real-time monitoring for the prevention of an epidemic. The possible future directions are:

- Real-time alert systems: Develop real-time alert systems that can notify healthcare providers and individuals about potential COVID-19 outbreaks or exposure risks. Utilize IoMT devices, mobile applications, and data analytics to deliver timely and accurate information, enabling rapid response and intervention measures.
- Privacy and security measures: Strengthen privacy and security measures to protect the sensitive healthcare data collected by IoMT devices. Implement robust encryption techniques, secure data storage, and access control mechanisms to ensure patient confidentiality. Address privacy concerns and regulatory compliance issues related to the collection, storage, and sharing of healthcare data.

### Funding
The funding of this work was provided by Princess Nourah bint Abdulrahman University Researchers Supporting Project number (PNURSP2023R410), Princess Nourah bint Abdulrahman University, Riyadh, Saudi Arabia. The funders participated in data curation, formal analysis and investigation part of the paper. The funders had no role in study design, decision to publish, or preparation of the manuscript.

### Grant Disclosures
The following grant information was disclosed by the authors:
Princess Nourah bint Abdulrahman University Researchers Supporting Project number: PNURSP2023R410.
Princess Nourah bint Abdulrahman University, Riyadh, Saudi Arabia.

### Competing Interests
Imran Ashraf is an Academic Editor with PeerJ Computer Science.

### Author Contributions
- Nouf Abdullah Almujally conceived and designed the experiments, analyzed the data, prepared figures and/or tables, and approved the final draft.
- Turki Aljrees conceived and designed the experiments, analyzed the data, prepared figures and/or tables, and approved the final draft.
- Muhammad Umer conceived and designed the experiments, performed the computation work, prepared figures and/or tables, and approved the final draft.

- Oumaima Saidani conceived and designed the experiments, performed the computation work, prepared figures and/or tables, and approved the final draft.
- Danial Hanif performed the experiments, performed the computation work, authored or reviewed drafts of the article, and approved the final draft.
- Nihal Abuzinadah performed the experiments, performed the computation work, authored or reviewed drafts of the article, and approved the final draft.
- Khaled Alnowaiser performed the experiments, analyzed the data, authored or reviewed drafts of the article, and approved the final draft.
- Imran Ashraf performed the experiments, analyzed the data, authored or reviewed drafts of the article, and approved the final draft.

### Data Availability

The dataset is available at Kaggle:

https://www.kaggle.com/iamhungundji/covid19-symptoms-checker

The implementation code is available at Zenodo:

MUmerSabir. (2023). MUmerSabir/IEEE-IOT: IoMT PeerJ CS (CodeFile). Zenodo. https://doi.org/10.5281/zenodo.7964918.

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

*Combating COVID-19—the role of robotics in managing public health and infectious diseases.* New York Avenue, United States: Science Robotics.