# Peer review of "IoMT based smart healthcare system to control outbreaks of the COVID-19 pandemic"

_PeerJ Computer Science, doi:10.7717/peerj-cs.1493_

## Round 0.1 · original submission · Major Revisions

The reviewers have provided comments, these comments must be addressed so the paper will be in very good standard for publication in PeerJ Computer Science.

·

Basic reporting

1. The abstract must be re-written focusing on the actual research results received.
2. The introduction is too short and does not explain the main research problem being solved in
the present work.
3. The main research contribution must be elaborated in the introduction section.
4. The literature review is weak, authors are suggested to incorporate the following works:
a. Sujath, R. A. A., Chatterjee, J. M., & Hassanien, A. E. (2020). A machine learning forecasting model for COVID-19 pandemic in India. Stochastic Environmental Research and Risk Assessment, 34, 959-972.
b. Yadav, S., Gulia, P., Gill, N. S., & Chatterjee, J. M. (2022). A real-time crowd monitoring and management system for social distance classification and healthcare using deep learning. Journal of Healthcare Engineering, 2022.

Experimental design

1. Figure 2-11 citation missing.
2. All the parameters used in the equations must be elaborated in the text.

Validity of the findings

1. Algorithm must be presented with the expected output.
2. The proposed work must be compared with state-of-the-art works from 2022/2023.

Additional comments

1. A thorough proofreading of the document is suggested.
2. 'et al.' must be avoided in the references.

Reviewer 2 ·

Basic reporting

This paper presents a road map for tracing the COVID-19 pandemic through the Internet of Things. The authors have also presented a near-real-time solution to monitor the affected patients. They have deployed a test bed for experimentation using the IoT infrastructure, quarantine center and a data warehouse for data analysis. The overall system seems good and could benefit any future covid like pandemic. The paper is easy to read and follow; however, it can be further improved by the following comments incorporations.
a) In the abstract, it is better to write some numeric results instead of qualitative results e.g. “Promising results are obtained using random forest and extra tree classifiers”. Could be changed to some quantitative results.
b) The validity of findings could be further strengthened by taking the consents of the domain experts and the system can be further improved with their input.
c) More latest literature review can be added to further explore the existing systems and making justification of the current framework
d) Pl. place section numbers to the heading like “COVID PHASES AND USAGE OF IOT DEVICES”
e) The proposed frameworks Figure 12 needs clarity as its blur at the moment and hard to read. Although the figure is self-explaining but however, it can be further transparent.
f) The results seem to be fine and satisfactory.
g) The language of the paper can be further improved to make it more understandable.

Experimental design

See above

Validity of the findings

See above

Additional comments

See above

Cite this review as

---

## Round 0.2 · Minor Revisions

Comments are provided by the reviewer and I do agree with the comments provided

·

Basic reporting

The paper is updated as per comments and some minor changes are required.

Experimental design

Algorithm 1 lacks expected outcome details.

Validity of the findings

In Table 3, the authors must compare the proposed work with some state-of-the-art works from 2022/2023.

Additional comments

A thorough proofreading of the document is suggested.
The possible future work and conclusion lack detailed elaboration.

Reviewer 2 ·

Basic reporting

The authors has addressed my review comments, therefore I am satisfied with the current version of the paper

Experimental design

The authors has addressed my review comments, therefore I am satisfied with the current version of the paper

Validity of the findings

The authors has addressed my review comments, therefore I am satisfied with the current version of the paper

Additional comments

No more comments

Cite this review as

---

## Round 0.3 · accepted · Accept

All of the comments have been addressed by the authors.